# Validation of a Compact and Self-Contained Pyrosequencing Platform for Clinical Screening of *RAS* Mutations in Thyroid Cancers

**DOI:** 10.3390/diagnostics15030390

**Published:** 2025-02-06

**Authors:** Anne Burkhardt, Chelsey L. Smith, Rajesh R. Singh

**Affiliations:** Department of Molecular Oncology, Quest Diagnostics Nichols Institute, Chantilly, VA 20151, USA; anne.x.burkhardt@questdiagnostics.com (A.B.); chelsey.l.smith@questdiagnostics.com (C.L.S.)

**Keywords:** pyrosequencing, Pyromark Q48, thyroid cancers, *RAS* genes, next-generation sequencing

## Abstract

**Background and Objectives:** Accurate screening of clinically significant tumor mutations is critical for precision medicine in oncology. This requires genotyping platforms with high accuracy and compatibility with varying DNA yields from challenging sample types. Here, we have validated a new, improved, compact, and self-contained pyrosequencing platform (Pyromark Q48 Autoprep; Q48) for screening *N*-, *K*- and *H-RAS* mutations in thyroid cancers. **Methods:** A set of 73 thyroid cancer and 16 non-thyroid cancer samples (fine needle aspirates and formalin-fixed paraffin-embedded) with known mutation status of *RAS* genes were tested using the Q48 platform. Performance parameters such as accuracy, precision, and limit-of-detection were established. Q48 workflow was compared to an older Q96 pyrosequencing platform to highlight the differences and advantages. *RAS* testing by pyrosequencing was also compared to a clinically validated next-generation sequencing platform using 56 thyroid cancer samples. **Results:** The Q48 Pyromark was found to be a very reliable platform suited for quick testing of *RAS* genes with complete accuracy, high precision, and high detection sensitivity. It had comparable accuracy, with higher sequencing success rates than NGS. The hands-on time, workflow ease, and efficiency were also significantly improved in comparison with the Q96 platform. **Conclusions:** Overall, the Q48 platform was found to be a well-suited and agile clinical sequencing platform to rapidly screen *RAS* mutations.

## 1. Introduction

Thyroid cancers represent the most common of the endocrine cancers [1]. The role of *RAS* oncogenes (*KRAS*, *NRAS*, and *HRAS*) in thyroid tumorigenesis has been well established. They also play an important role as diagnostic markers, specifically in cases where the malignancy is not apparent with morphological evaluation [2]. Mutations in *RAS* genes have been found to have a high positive predictive value for thyroid malignancy (overall 66%), where the mutated status of *HRAS* exhibited the highest positive predictive value (74%), followed by *NRAS* (65%) and *KRAS* at (51%) [3]. Furthermore, *RAS* gene mutations also occur more frequently in follicular thyroid cancer (35–40%) and rarely in classical papillary thyroid cancers, which also makes them important in the identification of the cancer sub-types. Although the diagnostic significance of *RAS* mutations in thyroid cancers is established, they have also been infrequently detected in benign thyroid nodules. This warrants for testing of additional markers such as *BRAF* mutations, *RET/PTC*, and *PAX8-PPAR**γ*** gene fusion to maximize diagnostic utility. The presence of genomic alterations in this panel of *RAS* and the additional markers strongly correlates with histologically confirmed malignancy (89–97%) [4]. In addition to the mutated status, studies have also shown that the presence of specific mutations of *RAS* genes has distinct clinicopathological outcomes. For instance, thyroid cancers which had mutations in codons 12 or 13 of *KRAS* were found to be associated with comparatively lower malignant outcomes (41.7%) as compared to mutations in codon 61 of *NRAS* (86.8%) and codon 61 of *HRAS* (95.5%). The mutational pattern among the *RAS* genes was also found to be more prevalent in poorly differentiated thyroid carcinoma types [5]. Consequently, it is critical to have accurate, rapid, and targeted sequencing of the three *RAS* genes in thyroid cancers. As most thyroid biopsy specimens are fine needle aspirate (FNA) cytology and formalin-fixed paraffin-embedded (FFPE) specimens, the routine screening methodology should also be compatible with compromised quantity and quality from these specimens.

Pyrosequencing is a sequencing or genotyping technology that uses the synthesis of a complementary DNA strand coupled with a light reaction for rapid genotyping of short stretches of genomic regions. More specifically, it includes the synthesis of a DNA strand using the template strand with a genomic region of interest and building a complementary strand by introducing one nucleotide at a time in a pre-determined order. The flow of a complementary nucleotide results in the incorporation of the nucleotide accompanied with the release of a pyrophosphate (PPi) which is converted to ATP by a subsequent enzymatic reaction. The generated ATP is then used as a co-factor of the enzyme luciferase to oxidize luciferin to oxyluciferin with the associated emission of light. The detection of the emitted light indicates the incorporation of a nucleotide, and the intensity of the light is proportional to the number of nucleotides incorporated [6].

The ability of rapid, focused, and semi-quantitative genotyping by pyrosequencing at a relatively low cost has made it a preferred platform for a variety of applications such as screening of polymorphisms, short sequence repeats, insertions and deletions, copy number variation and allelic balance, and methylation. Consequently, it has found acceptance in wide-ranging fields like forensics, hereditary screening, microbiology, and oncology [7,8,9,10,11]. The most widely used platform for pyrosequencing historically has been the Pyromark Q96 platform (Qiagen) [12,13,14,15]. The instrumentation of the Pyromark Q96 platform has two major components, namely the Q96 vacuum workstation and the Q96 ID along with a computer to run the Q96 software program. Recently, an improved, more compact, and self-contained pyrosequencing platform the Pyromark Q48 Autoprep (referred to as Q48 henceforth) was launched by Qiagen. This platform has a drastically different build design as compared to the Q96 Pyromark with all wet-lab components necessary for the pyrosequencing assay contained in a compact unit.

Our clinical molecular laboratory has validated and implemented a pyrosequencing assay for screening of *RAS* genes (*KRAS*, *NRAS*, and *HRAS*) in thyroid tumors for frequently occurring oncogenic mutations (in exons 2 and 3) using an older Q96 Pyromark platform. As FNA and FFPE are the most frequent specimen types for thyroid tumors, the pyrosequencing platform provides a suitable PCR-based platform that is feasible for these samples which routinely yield low quantity and quality of DNA. The current study involves the validation of the new Q48 platform for its utility as a replacement and an alternative platform for *RAS* testing. For this purpose, 89 samples with known mutation status of the three *RAS* genes either by pyrosequencing using Q96, Sanger sequencing, or next-generation sequencing (NGS) were tested using the Q48 platform and the concordance tested to establish the accuracy of mutation detection. Testing and establishment of additional performance parameters such as precision and limit-of-detection (LOD) were also established.

## 2. Materials and Methods

### 2.1. Study Design

A total of 89 tumor samples (22 FFPE and 67 FNA samples; 73 thyroid and 16 non-thyroid cancers) were used in the study. The non-thyroid samples included six colorectal, four lung, three lymph node biopsies, two liver, and one peritoneal biopsy. These samples had previously been evaluated for mutations in the most mutated exons of RAS genes (*K*-, *N*-, and *H*- *RAS*; exons 2 and 3) by sequencing via pyrosequencing on Q96 Pyromark, Sanger sequencing, or NGS. The samples were tested using the Q48 pyrosequencing platform and the ability to detect the mutations was assessed. Test performance parameters to validate the platform for clinical testing such as accuracy, precision, and limit-of-detection were also established.

The accuracy of the Q48 Pyromark platform was established by testing the sample set mentioned above. These were selected for their known status of *RAS* mutation as established by prior testing on validated clinical platforms in our laboratory. Accuracy was established by examining the result concordance regarding the presence or absence of mutations along with the identity of the mutation detected. The precision of testing was established by repeatedly testing positive and negative samples in triplicates within the same run (intra-run precision) and on three independent runs (inter-run precision), and the results were compared. The lower limit-of-detection (LOD) for each of the *RAS* genes was established by testing samples with high to low levels of mutations. These were generated by sequentially diluting the DNA of a positive sample in the background of DNA from a negative sample to generate progressively decreasing mutation levels. The DNA of the positive and negative samples was individually diluted to a stock concentration of 5 ng/µL and subsequently mixed as needed to obtain sequentially decreasing mutation levels (final DNA concentration of 5 ng/µL). It must also be noted that all the analytical validation studies were also performed in parallel on two Q48 instruments to establish inter-instrument reproducibility of testing.

To test the versatility of NGS testing to sequence DNA with low quality and yields from FFPE and FNA samples with consistent success, a set of 56 thyroid samples (48 FNA and 8 FFPE), which was tested on the pyrosequencing platforms (Q48 or Q96 platforms), was also tested using an NGS assay to screen for mutations in the three *RAS* genes.

### 2.2. DNA Extraction

DNA (total nucleic acid) from FFPE and FNA thyroid samples was extracted using the FormaPure kit (Agentcourt Bioscience Corp, Beverley, MA, USA). As the *RAS* testing by pyrosequencing and NGS platforms uses DNA only, the nucleic acid input for thyroid samples will be referred to as ‘DNA’ henceforth. DNA extraction from non-thyroid FFPE samples was performed using the QIAamp DNA FFPE Tissue Kit (Qiagen, Germantown, MD, USA).

### 2.3. Pyrosequencing

The genomic regions of interest in the three *RAS* genes encompassing exons 2 and 3 were amplified using sequence-specific primers. Pyrosequencing of the amplified genomic regions was performed using the Pyromark Q48 Advanced reagents kit (Qiagen), and Pyromark Q48 Magnetic beads (Qiagen) for use on the Pyromark Q48 instruments as per the manufacturer’s instructions.

### 2.4. Sequencing of RAS Genes in Thyroid Samples Using NGS

DNA from a set of 56 thyroid samples (48 FNAs and 8 FFPEs) that was tested for *RAS* mutations using Q96 or Q48 Pyrosequencers was also screened using a clinically validated NGS platform for *RAS* mutations. The NGS platform employed hybridization probe-based enrichment to capture genomic regions of interest target enrichment using custom-designed probes for *K*-, *N*-, and *H*-*RAS* genes (Integrated DNA Technologies, Coralville, IA, USA; IDT). Libraries were prepared using the NEBNext FFPE DNA Repair Module v2 and NEBNext Ultra II FS DNA Library Prep Kit for Illumina (New England Biolabs, Ipswich, MA, USA). Library preparation was followed by sequencing on the NextSeq500 sequencer (Illumina, San Diego, CA, USA).

For the processing of sequencing results, Illumina’s bcl2fastq software v2 was used for de-multiplexing and conversion of the NextSeq500 BCL files. The BWA alignment package was used to align the FASTQ files to the Genome Reference Consortium human genome build 37 (GRCh37). The sorting and indexing of reads were performed by SAMtools and subsequent read duplications were removed by Picard Tools (Broad Institute, Cambridge, MA, USA). The Genome Analysis Toolkit (GATK) was used to perform the local realignment and base quality score recalibration [16]. Mapped reads were filtered using a mapping score of ≥30 and a base quality score of ≥20 before being analyzed. SAMtools Pysam was used to calculate the average minimum depth of coverage for all regions of interest (ROI). MuTect2 [17] and LoFreq [18] were used to make calls on SNVs and short indels.

## 3. Results

### 3.1. Performance of the Q48 Pyromark for Clinical Testing of RAS Genes

In order to establish the new Q48 Pyromark as a viable clinical platform to screen *RAS* genes for mutations, its ability to identify the mutations on pre-tested samples was assessed. DNA from a total of 89 samples with known mutation status of the three *RAS* genes as tested on multiple validated platforms in the laboratory were tested on the Q48 platform and the concordance was assessed. Several parameters gauging the performance of *RAS* testing on the Q48 platform such as accuracy, precision, and limit-of-detection were established for each of the *RAS* genes and are described below.

#### 3.1.1. *KRAS* Mutation Screening

Accuracy of *KRAS* mutation screening: To test the accuracy of screening *KRAS*, DNA from 29 pre-tested samples (15 FNAs and 14 FFPEs, 17 thyroid and 12 non-thyroid cancers) with known *KRAS* mutation status were tested on the two Q48 platforms in the laboratory (Q48 #1 and Q48 #2) and results were compared. The sample set consisted of 13 positive and 16 negative samples. Both Q48 instruments were able to detect each of the expected *KRAS* mutations in the set of positives and no mutations were detected in the negative samples indicating complete concordance, which indicated a high accuracy of *KRAS* mutation detection. The results are summarized in Table 1 below and representative pyrograms for *KRAS* exon 2 and 3 mutations as detected by Q48 are shown in Figure 1. Non-thyroid samples are indicated by * in the table.Precision for *KRAS* mutation detection: To establish the precision of *KRAS* mutation testing, a total of six samples (three positives and three negatives; as detected by prior testing on validated other platforms) were tested thrice on distinct runs (inter-run) and within the same run (intra-run) on both the Q48 instruments and compared. Results showed that the mutation status and the identity of the mutation in both inter-run or intra-run repeats were consistently detected accurately on both instruments indicating excellent precision. Results have been summarized in Table 2 and Table 3 below.*KRAS* limit-of-detection (LOD): Limit-of-detection (LOD) for *KRAS* was established by testing different dilutions of a positive DNA sample (*KRAS*, p.G12D) in the background of DNA from a negative sample to create mutations at different levels (66% to 4%). These dilutions were tested on both Q48 instruments and their ability to detect them was measured. Results showed that both the instruments were able to detect the mutation at all levels including the lowest variant allelic frequencies (VAFs) which established the LOD at around 4%. Results are summarized in Table 4.

#### 3.1.2. NRAS Mutation Screening

Accuracy of *NRAS* mutation screening: To test the accuracy of screening *NRAS*, DNA from 30 pre-tested samples (25 FNA and 5 FFPE; 26 thyroid and 4 non-thyroid samples) with known *NRAS* mutation status were tested on the two Q48 platforms in the lab and results compared. The sample set consisted of 17 positive and 13 negative samples. Both Q48 instruments were able to detect each of the expected *NRAS* mutations in the set of positives and no mutations were detected in the negative samples indicating complete concordance, which indicated a high accuracy of *NRAS* mutation detection. The results are summarized in Table 5. Representative results for *NRAS* positives and negatives detected by the Q48 Pyromark are shown in Figure 2. Non-thyroid samples are indicated by * in the table.Precision for *NRAS* mutation detection: To establish the precision of *NRAS* mutation testing a total of six samples (three positives and three negatives as identified by prior testing on validated platforms) were tested thrice on distinct runs (inter-run) and within the same run (intra-run) on both Q48 instruments and results compared. Results showed that the mutation status and the identity of the mutation in each inter-run and intra-run repeats were consistently detected on both instruments indicating excellent precision. Results are summarized in Table 6 and Table 7 below.*NRAS* limit-of-detection (LOD): Limit-of-detection (LOD) for *NRAS* was established by testing different dilutions of a positive DNA sample (*NRAS*, p.G12D) in the background of DNA from a negative sample to create mutations at different levels (41% to 5%). These dilutions were tested on both Q48 instruments and their ability to detect them was measured. Results showed that both instruments were able to detect the mutation at all levels including the lowest variant allelic frequencies (VAFs) which established the LOD at around 6%. Results are summarized in Table 8.

#### 3.1.3. HRAS Mutation Screening

Accuracy of *HRAS* mutation screening: To test the accuracy of screening *HRAS*, DNA from 30 pre-tested thyroid samples (27 FNA and 3 FFPE) with known *HRAS* mutation status were tested on the two Q48 platforms in the lab, and results were compared. The sample set consisted of 6 positive and 24 negative samples. Both Q48 instruments were able to detect each of the expected *HRAS* mutations in the set of positives and no mutations were detected in the negative samples indicating complete concordance, which indicated a high accuracy of *HRAS* mutation detection. The results are summarized in Table 9. Representative results are shown in Figure 3.Precision for *HRAS* mutation detection: To establish the precision or reproducibility of *HRAS* mutation testing a total of six samples (three positives and three negatives as identified by prior testing on validated platforms) were tested thrice on distinct runs (inter-run) and within the same run (intra-run) on both the Q48 instruments. Results showed that the mutation status and the identity of the mutation detected in each inter-run and intra-run repeats were concordant on both instruments indicating excellent precision. Results are summarized in Table 10 and Table 11 below.*HRAS* limit-of-detection (LOD): Limit-of-detection (LOD) for *HRAS* was established by testing different dilutions of a positive DNA sample (*HRAS*, p.G12D) in the background of DNA from a negative sample to create mutations at different levels (53% to 3.3%). These dilutions were tested on both Q48 instruments and their ability to detect them was measured. Results showed that both instruments were able to detect mutation at all levels including the lowest variant allelic frequencies (VAFs) which established the LOD at around 7%. Results are summarized in Table 12.

### 3.2. Performance Comparison of Pyrosequencing Versus NGS for RAS Testing in Thyroid Samples

To test the applicability of NGS for mutational screening of thyroid cancer samples and compare it to the pyrosequencing platforms, a set of 56 thyroid tumor samples (34 negative and 22 positive) previously tested on pyrosequencing platforms (Q96 or Q48) for mutations in *RAS* genes was also tested on the NGS platform. Out of the 56 samples (48 FNA and 8 FFPE), 12 samples (7 FNA and 5 FFPE) failed NGS testing due to sub-optimal sequencing and were unable to reach the minimum required sequencing depth of 300× (21.4% failure rate). Nine of these twelve samples completely failed sequencing in all the regions of *RAS* genes sequenced and three failed sequencing on one of the *RAS* exons. Notably, the failed samples had low DNA concentrations in general (with a total DNA input of <5 ng). This indicated that pyrosequencing platforms were more successful for *RAS* testing in thyroid samples as compared to NGS, specifically for challenging samples. In the remaining samples that worked on NGS (44 total; 24 negative and 20 positive), a high overall concordance was observed with regard to the mutation status and mutations detected in all the samples except one (Table 13; sample 31). In this sample, the mutation detected by NGS was a dual mutation involving the last nucleotide of *KRAS* codon 60 (resulting in a silent mutation) and the first nucleotide of codon 61 resulting in the Q61K mutation. As the mutation in codon 60 interferes with primer binding, this mutation was missed by the Q96 platform. Results are summarized in Table 13. Samples that failed NGS in all regions of *RAS* sequenced are indicated by the symbol #.

### 3.3. Q96 and Q48 Pyromarks: Instrumentation and Workflow Comparison

Although the fundamental principles and process of pyrosequencing used are the same in the Q96 and the Q48 instruments, their instrumentation, size or footprint, and the workflow involved are considerably different. The Q96 has two separate components, the Pyromark Q96, and an accompanying bulky vacuum station. The vacuum station is used to capture the Biotin-labeled PCR product (template for pyrosequencing) using streptavidin beads, which includes several washes. This is followed by converting the template to single-stranded DNA and annealing the sequencing primer to prepare it for pyrosequencing. The pyrosequencing is performed on the Q96 Pyromark, which is a light-proof housing with a light detection system to record the signal emitted from the pyrosequencing reaction. It also has an onboard computer for data analysis. The total footprint of the Q96 and the associated vacuum station is 0.492 m^2^. In comparison, the Q48 is a very compact pyrosequencer containing everything needed to purify and prepare the template DNA, and perform the pyrosequencing reaction and light detection, making it a self-sufficient system. It has a footprint of 0.141 m^2^, which is considerably less (<30%) as compared to Q96 Pyromark. Additionally, in Q96, the sequencing primer annealing and the pyrosequencing reaction are conducted in a 96-well plate. In comparison, in Q48 these steps occur in a uniquely designed disk with wells provided at the periphery into which reagents are dispensed from distinct chambers of a cartridge for the pyrosequencing reaction (Appendix A).

The overall pyrosequencing workflow is considerably manual in Q96 as compared to Q48 with more hands-on time and consumables needed. Specifically, the use of the vacuum station is a very manual process in Q96, which takes an additional 40 min of hands-on work in order to capture the Biotin-labeled PCR product (step indicated by a red asterisk in Appendix A), subject it to several washes and anneal it to the sequencing primer. This process is completely automated in the Q48 platform making it more efficient and abrogate errors (comparison summarized in Appendix A). Although Q48 had significant improvements over Q96 overall, it also had some limitations and minor challenges. Firstly, the throughput was 48 samples per run as compared to 96 samples on Q96 Pyromark. Also, the signal strength or pyrogram peak heights in the Q48 instrument were lower to a slight extent as compared to Q96. This required us to add PCR cycles to increase the levels of the PCR product which in turn helped increase the pyrosequencing signal. There was also a pattern of increasing noise or signal background after multiple runs, which was not prominent in Q96. The compacted design and associated changes in the fluidics of Q48 could be causative for this. A weekly pyrophosphatase cleaning to remove any lingering pyrophosphate in the instruments was helpful in minimizing this issue.

## 4. Discussion

The need for sequencing and genotyping platforms that can perform targeted sequencing of known mutational hotspots in genes of clinical significance is very valuable in oncology [19]. Additionally, it also becomes critical for these platforms to be compatible with testing DNA samples with low quality and quantity to work with routine tumor samples. Historically, PCR-based platforms that suit these requirements were predominantly used for screening tumors [20,21,22,23,24]. Pyrosequencing is one such platform that can provide accurate results with a short turn-around time.

Here, we have fully tested and validated the use of a new, re-designed, and compact pyrosequencing platform, Pyromark Q48, for its applicability as a routine platform to detect the mutational status of three *RAS* genes (*K*-, *N*-, and *H-RAS*) in thyroid cancers. The most commonly mutated genomic regions of the *RAS* genes (exons 2 and 3) were targeted and the ability of the Q48 platform to successfully sequence and identify the mutations was validated. A set of 89 tumor samples (73 thyroid and 16 non-thyroid) FFPE and FNA samples with known mutation status (36 positives and 53 negatives) was used to establish the sample-type compatibility and accuracy of *RAS* gene testing. The mutational status was known by prior testing on other validated clinical testing platforms in our laboratory (pyrosequencing by Q96 Pyromark, Sanger sequencing, and NGS). Q48 results showed that the platform accurately detected the *RAS* mutations on these samples and exhibited complete concordance with the other validated clinical platforms. Additionally, the Q48 platform showed a high degree of precision within sequencing runs (intra-run) and across sequencing runs (inter-run). We tested these performance parameters across two distinct Q48 instruments and the same levels of accuracy and precision were observed across them indicating a high level of inter-instrument reproducibility, which is a very desirable consistency in a clinical testing platform. Limit-of-detection studies were also performed for each *RAS* gene using diluted positive samples and the LOD was established to be around 5% mutated sequence in the background of wildtype, which is the expected detection sensitivity for pyrosequencing platforms.

In recent years, next-generation sequencing technologies have been preferably used for clinical sequencing of tumors and have been considered to be viable alternatives for the smaller, PCR-based single gene and low-scale sequencing platforms such as pyrosequencing, Sanger sequencing, and real-time PCR technologies [25,26,27,28,29]. Although their suitability is justified for screening multiple markers simultaneously in a tumor sample, the value and application of the single gene screening technologies have not been completely obviated. These lower throughput PCR-based technologies still hold relevance, especially for testing samples with relatively scarce DNA yield and highly variable quality. This was evident in our study where a clinically validated NGS platform in our laboratory was tested to screen *RAS* genes in the FFPE and FNA thyroid samples. Although NGS was able to successfully test the samples with high concordance to pyrosequencing, testing on a subset of samples failed (~22%) due to the low amount of DNA present in the samples. However, these samples were successfully tested by clinically validated pyrosequencing platforms. It must be noted that the majority of the samples which failed NGS had very low amounts of DNA yields, which is a characteristic limitation of thyroid needle biopsy samples. Consequently, this subset did not undergo successful testing by NGS despite including DNA repair steps prior to hybrid capture for NGS and sequencing conditions to provide adequate sequencing opportunity (enough to achieve an average sequencing depth of ~3000x across all *RAS* exons). This exhibits the vulnerability of the NGS technology when encountered with a wide variation in sample availability and quality such as thyroid samples. In comparison, the pyrosequencing platforms also provide a quicker turn-around time of 2–3 days as compared to 5–7 days by NGS indicating them to be a quicker, cost-effective, and robust option to screen *RAS* genes in challenging tumor samples.

## 5. Conclusions

The compact and self-contained Pyromark Q48 Autoprep platform is a well-suited clinical sequencing platform for screening targeted clinically relevant genes such as the *RAS* genes in challenging sample types like FFPE and FNA specimens of thyroid cancers. It exhibited high accuracy and precision in mutation detection. The sequencing success rate was also considerably higher as compared to the NGS platform, making it very reliable and suitable for routine screening of targeted genomic regions.

## Figures and Tables

**Figure 1 diagnostics-15-00390-f001:**
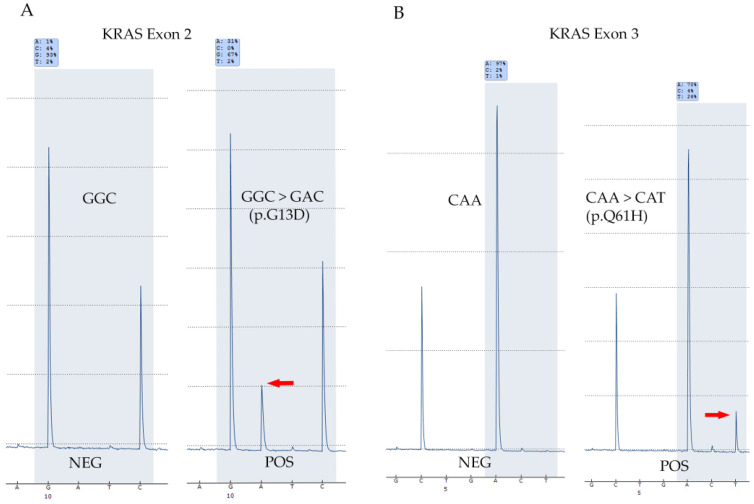
*KRAS* mutation detection by Q48 Pyromark. (**A**) Representative pyrograms showing the results for a negative and positive case in *KRAS* exon 2. (**B**) Examples of *KRAS* exon 3 negative and positive cases as detected by Q48 Pyromark. Peaks indicating the mutant sequence are marked by red arrows. NEG: negative; and POS: positive.

**Figure 2 diagnostics-15-00390-f002:**
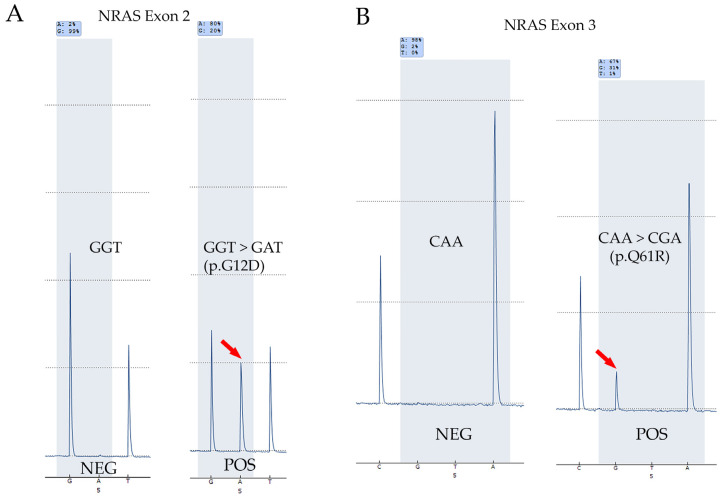
*NRAS* mutation detection by Q48 Pyromark. (**A**) Representative pyrograms for a negative and positive case in *NRAS* exon 2. (**B**) Examples of *NRAS* exon 3 negative and positive cases as detected by Q48 Pyromark. Peaks indicating the mutant sequence are marked by red arrows. NEG: negative; and POS: positive.

**Figure 3 diagnostics-15-00390-f003:**
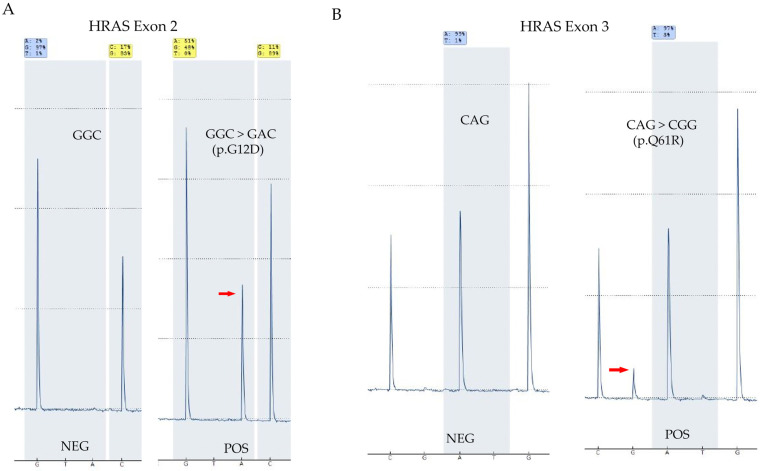
*HRAS* mutation detection by Q48 Pyromark. (**A**) Representative results for a negative and positive case for *HRAS* exon 2. (**B**) Examples of *HRAS* exon 3 negative and positive cases. Peaks indicating the mutant sequence are marked by red arrows. NEG: negative; and POS: positive.

**Table 1 diagnostics-15-00390-t001:** Accuracy of *KRAS* mutation detection by Q48 platforms.

Sample No.	Sample Type	Original Testing Platform	Gene	Expected Result	Result (Q48 #1)	Result (Q48 #2)
* 1	FFPE	Q96 Pyromark	*KRAS*	POS, G12D	POS, G12D	POS, G12D
* 2	FFPE	Q96 Pyromark	*KRAS*	POS, Q61L	POS, Q61L	POS, Q61L
* 3	FFPE	Q96 Pyromark	*KRAS*	POS, G12D	POS, G12D	POS, G12D
* 4	FFPE	Q96 Pyromark	*KRAS*	POS, G12C	POS, G12C	POS, G12C
* 5	FFPE	Q96 Pyromark	*KRAS*	POS, Q61L	POS, Q61L	POS, Q61L
* 6	FFPE	Q96 Pyromark	*KRAS*	POS, G12D	POS, G12D	POS, G12D
* 7	FFPE	Q96 Pyromark	*KRAS*	POS, G12D	POS, G12D	POS, G12D
* 8	FFPE	Q96 Pyromark	*KRAS*	POS, G12C	POS, G12C	POS, G12C
* 9	FFPE	Q96 Pyromark	*KRAS*	POS, G12C	POS, G12C	POS, G12C
* 10	FFPE	Q96 Pyromark	*KRAS*	POS, G12C	POS, G12C	POS, G12C
* 11	FFPE	Sanger sequencing	*KRAS*	POS, G12D	POS, G12D	POS, G12D
12	FNA	Q96 Pyromark	*KRAS*	POS, Q61R	POS, Q61R	POS, Q61R
* 13	FFPE	NGS	*KRAS*	POS, G12V	POS, G12V	POS, G12V
14	FNA	Q96 Pyromark	*KRAS*	NEG	NEG	NEG
15	FNA	Q96 Pyromark	*KRAS*	NEG	NEG	NEG
16	FNA	Q96 Pyromark	*KRAS*	NEG	NEG	NEG
17	FNA	Q96 Pyromark	*KRAS*	NEG	NEG	NEG
18	FNA	Q96 Pyromark	*KRAS*	NEG	NEG	NEG
19	FFPE	Q96 Pyromark	*KRAS*	NEG	NEG	NEG
20	FFPE	Q96 Pyromark	*KRAS*	NEG	NEG	NEG
21	FNA	Q96 Pyromark	*KRAS*	NEG	NEG	NEG
22	FNA	Q96 Pyromark	*KRAS*	NEG	NEG	NEG
23	FNA	Q96 Pyromark	*KRAS*	NEG	NEG	NEG
24	FNA	Q96 Pyromark	*KRAS*	NEG	NEG	NEG
25	FNA	Q96 Pyromark	*KRAS*	NEG	NEG	NEG
26	FNA	Q96 Pyromark	*KRAS*	NEG	NEG	NEG
27	FNA	Q96 Pyromark	*KRAS*	NEG	NEG	NEG
28	FNA	Q96 Pyromark	*KRAS*	NEG	NEG	NEG
29	FNA	Q96 Pyromark	*KRAS*	NEG	NEG	NEG

* Non-thyroid samples.

**Table 2 diagnostics-15-00390-t002:** Intra-run precision of *KRAS* mutation detection by Q48 platforms.

Samples	Sample ID	Expected *KRAS* Mutation	Mutation Frequency (Expected %)	Detected Mutation (Q48 #1)	Mutation Frequency (%) (Q48 #1)	Detected Mutation (Q48 #2)	Mutation Frequency (%) (Q48 #2)
1	1	G12D	12	G12D	16	G12D	16
	1	G12D	12	G12D	17	G12D	17
	1	G12D	12	G12D	16	G12D	17
2	2	G12D	25	G12D	30	G12D	30
	2	G12D	25	G12D	29	G12D	28
	2	G12D	25	G12D	30	G12D	29
3	3	G12D	65	G12D	64	G12D	66
	3	G12D	65	G12D	62	G12D	61
	3	G12D	65	G12D	64	G12D	64
4	4	NEG	N/A	NEG	N/A	NEG	N/A
	4	NEG	N/A	NEG	N/A	NEG	N/A
	4	NEG	N/A	NEG	N/A	NEG	N/A
5	5	NEG	N/A	NEG	N/A	NEG	N/A
	5	NEG	N/A	NEG	N/A	NEG	N/A
	5	NEG	N/A	NEG	N/A	NEG	N/A
6	6	NEG	N/A	NEG	N/A	NEG	N/A
	6	NEG	N/A	NEG	N/A	NEG	N/A
	6	NEG	N/A	NEG	N/A	NEG	N/A

N/A: Not Applicable as the samples are negative.

**Table 3 diagnostics-15-00390-t003:** Inter-run precision of *KRAS* mutation detection by Q48 platforms.

Run No.	Sample ID	Expected *KRAS* Mutation	Mutation Frequency (%) (Expected)	Detected (Q48 #1)	Mutation Frequency (%) (Q48 #1)	Detected (Q48 #2)	Mutation Frequency (%) (Q48 #2)
1	1	G12D	12	G12D	16	G12D	16
2	1	G12D	12	G12D	18	G12D	21
3	1	G12D	12	G12D	16	G12D	16
1	2	G12D	25	G12D	29	G12D	28
2	2	G12D	25	G12D	31	G12D	32
3	2	G12D	25	G12D	28	G12D	29
1	3	G12D	65	G12D	66	G12D	64
2	3	G12D	65	G12D	66	G12D	69
3	3	G12D	65	G12D	64	G12D	66
1	4	NEG	N/A	NEG	N/A	NEG	N/A
2	4	NEG	N/A	NEG	N/A	NEG	N/A
3	4	NEG	N/A	NEG	N/A	NEG	N/A
1	5	NEG	N/A	NEG	N/A	NEG	N/A
2	5	NEG	N/A	NEG	N/A	NEG	N/A
3	5	NEG	N/A	NEG	N/A	NEG	N/A
1	6	NEG	N/A	NEG	N/A	NEG	N/A
2	6	NEG	N/A	NEG	N/A	NEG	N/A
3	6	NEG	N/A	NEG	N/A	NEG	N/A

N/A: Not Applicable as the samples are negative.

**Table 4 diagnostics-15-00390-t004:** Limit-of-detection (LOD) for *KRAS* mutation detection by Q48 platforms.

Sample Dilution Levels	Expected Mutation (%)	Detected Mutation (%) (Q48 #1)	Detected Mutation (%) (Q48 #2)
(Undiluted) 100%	66	64	61
50%	33	31	31
25%	16.50	21	19
12.50%	8.25	9	8
6.25%	4	5	4

**Table 5 diagnostics-15-00390-t005:** Accuracy of *NRAS* mutation detection by Q48 platforms.

Sample No.	Sample Type	Original Testing Platform	Gene	Expected Result	Result (Q48 #1)	Result (Q48 #2)
1	FNA	Q96 Pyromark	*NRAS*	POS, Q61R	POS, Q61R	POS, Q61R
2	FNA	Q96 Pyromark	*NRAS*	POS, Q61R	POS, Q61R	POS, Q61R
3	FNA	Q96 Pyromark	*NRAS*	POS, Q61K	POS, Q61K	POS, Q61K
4	FNA	Q96 Pyromark	*NRAS*	POS, Q61R	POS, Q61R	POS, Q61R
5	FNA	Q96 Pyromark	*NRAS*	POS, Q61R	POS, Q61R	POS, Q61R
6	FNA	Q96 Pyromark	*NRAS*	POS, Q61R	POS, Q61R	POS, Q61R
7	FNA	Q96 Pyromark	*NRAS*	POS, Q61R	POS, Q61R	POS, Q61R
8	FNA	Q96 Pyromark	*NRAS*	POS, Q61K	POS, Q61K	POS, Q61K
9	FNA	Q96 Pyromark	*NRAS*	POS, G13R	POS, G13R	POS, G13R
10	FNA	Q96 Pyromark	*NRAS*	POS, Q61K	POS, Q61K	POS, Q61K
* 11	FFPE	Sanger sequencing	*NRAS*	POS, G12D	POS G12D	POS G12D
* 12	FFPE	Sanger sequencing	*NRAS*	POS, G12D	POS, G12D	POS, G12D
* 13	FFPE	Sanger sequencing	*NRAS*	POS, G12D	POS, G12D	POS, G12D
* 14	FFPE	Sanger sequencing	*NRAS*	POS, G12D	POS G12D	POS G12D
15	FNA	Q96 Pyromark	*NRAS*	POS, Q61R	POS, Q61R	POS, Q61R
16	FNA	Q96 Pyromark	*NRAS*	POS, Q61R	POS, Q61R	POS, Q61R
17	FNA	Q96 Pyromark	*NRAS*	POS, Q61R	POS, Q61R	POS, Q61R
18	FNA	Q96 Pyromark	*NRAS*	NEG	NEG	NEG
19	FNA	Q96 Pyromark	*NRAS*	NEG	NEG	NEG
20	FFPE	Q96 Pyromark	*NRAS*	NEG	NEG	NEG
21	FNA	Q96 Pyromark	*NRAS*	NEG	NEG	NEG
22	FNA	Q96 Pyromark	*NRAS*	NEG	NEG	NEG
23	FNA	Q96 Pyromark	*NRAS*	NEG	NEG	NEG
24	FNA	Q96 Pyromark	*NRAS*	NEG	NEG	NEG
25	FNA	Q96 Pyromark	*NRAS*	NEG	NEG	NEG
26	FNA	Q96 Pyromark	*NRAS*	NEG	NEG	NEG
27	FNA	Q96 Pyromark	*NRAS*	NEG	NEG	NEG
28	FNA	Q96 Pyromark	*NRAS*	NEG	NEG	NEG
29	FNA	Q96 Pyromark	*NRAS*	NEG	NEG	NEG
30	FNA	Q96 Pyromark	*NRAS*	NEG	NEG	NEG

* Non-thyroid samples.

**Table 6 diagnostics-15-00390-t006:** Intra-run precision of *NRAS* mutation detection by Q48 platforms.

Samples	Sample ID	Expected *NRAS* Mutation	Mutation Frequency (Expected %)	Detected Mutation (Q48 #1)	Mutation Frequency (%) (Q48 #1)	Detected Mutation (Q48 #2)	Mutation Frequency (%) (Q48 #2)
1	1	G12D	51	G12D	51	G12D	53
	1	G12D	51	G12D	52	G12D	51
	1	G12D	51	G12D	53	G12D	49
2	2	G12D	12	G12D	15	G12D	14
	2	G12D	12	G12D	14	G12D	14
	2	G12D	12	G12D	14	G12D	14
3	3	G12D	36	G12D	37	G12D	36
	3	G12D	36	G12D	37	G12D	36
	3	G12D	36	G12D	36	G12D	38
4	4	NEG	N/A	NEG	N/A	NEG	N/A
	4	NEG	N/A	NEG	N/A	NEG	N/A
	4	NEG	N/A	NEG	N/A	NEG	N/A
5	5	NEG	N/A	NEG	N/A	NEG	N/A
	5	NEG	N/A	NEG	N/A	NEG	N/A
	5	NEG	N/A	NEG	N/A	NEG	N/A
6	6	NEG	N/A	NEG	N/A	NEG	N/A
	6	NEG	N/A	NEG	N/A	NEG	N/A
	6	NEG	N/A	NEG	N/A	NEG	N/A

N/A: Not Applicable as the samples are negative.

**Table 7 diagnostics-15-00390-t007:** Inter-run precision of *NRAS* mutation detection by Q48 platforms.

Run No	Sample ID	Expected *NRAS* Mutation	Mutation Frequency (Expected %)	Detected Mutation (Q48 #1)	Mutation Frequency (%) (Q48 #1)	Detected Mutation (Q48 #2)	Mutation Frequency (%) (Q48 #2)
1	1	G12D	51	G12D	51	G12D	50
2	1	G12D	51	G12D	56	G12D	59
3	1	G12D	51	G12D	52	G12D	53
1	2	G12D	12	G12D	12	G12D	12
2	2	G12D	12	G12D	13	G12D	15
3	2	G12D	12	G12D	11	G12D	12
1	3	G12D	36	G12D	36	G12D	35
2	3	G12D	36	G12D	39	G12D	41
3	3	G12D	36	G12D	34	G12D	36
1	4	NEG	N/A	NEG	N/A	NEG	N/A
2	4	NEG	N/A	NEG	N/A	NEG	N/A
3	4	NEG	N/A	NEG	N/A	NEG	N/A
1	5	NEG	N/A	NEG	N/A	NEG	N/A
2	5	NEG	N/A	NEG	N/A	NEG	N/A
3	5	NEG	N/A	NEG	N/A	NEG	N/A
1	6	NEG	N/A	NEG	N/A	NEG	N/A
2	6	NEG	N/A	NEG	N/A	NEG	N/A
3	6	NEG	N/A	NEG	N/A	NEG	N/A

N/A: Not Applicable as the samples are negative.

**Table 8 diagnostics-15-00390-t008:** Limit-of-detection (LOD) for *NRAS* mutation detection by Q48 platforms.

Sample Dilution Levels	Expected *NRAS* Mutation (%)	Detected Mutation (%) (Q48 #1)	Detected Mutation (%) (Q48 #2)
Undiluted (100%)	41	45	43
50%	20.5	26	26
25%	10.25	15	15
12.50%	5.13	6	6

**Table 9 diagnostics-15-00390-t009:** Accuracy of *HRAS* mutation detection by Q48 platforms.

Sample No.	Sample Type	Original Testing Platform	Gene	Expected Result	Result (Q48 #1)	Result (Q48 #1)
1	FNA	Q96 Pyromark	*HRAS*	POS, Q61R	POS Q61R	POS Q61R
2	FNA	Q96 Pyromark	*HRAS*	POS, Q61R	POS Q61R	POS Q61R
3	FNA	Q96 Pyromark	*HRAS*	POS, Q61R	POS Q61R	POS Q61R
4	FNA	Q96 Pyromark	*HRAS*	POS, Q61R	POS, Q61R	POS, Q61R
5	FFPE	Q96 Pyromark	*HRAS*	POS, Q61R	POS, Q61R	POS, Q61R
6	FNA	Q96 Pyromark	*HRAS*	POS, Q61R	POS, Q61R	POS, Q61R
7	FFPE	Q96 Pyromark	*HRAS*	NEG	NEG	NEG
8	FNA	Q96 Pyromark	*HRAS*	NEG	NEG	NEG
9	FNA	Q96 Pyromark	*HRAS*	NEG	NEG	NEG
10	FNA	Q96 Pyromark	*HRAS*	NEG	NEG	NEG
11	FNA	Q96 Pyromark	*HRAS*	NEG	NEG	NEG
12	FNA	Q96 Pyromark	*HRAS*	NEG	NEG	NEG
13	FNA	Q96 Pyromark	*HRAS*	NEG	NEG	NEG
14	FNA	Q96 Pyromark	*HRAS*	NEG	NEG	NEG
15	FNA	Q96 Pyromark	*HRAS*	NEG	NEG	NEG
16	FNA	Q96 Pyromark	*HRAS*	NEG	NEG	NEG
17	FNA	Q96 Pyromark	*HRAS*	NEG	NEG	NEG
18	FNA	Q96 Pyromark	*HRAS*	NEG	NEG	NEG
19	FNA	Q96 Pyromark	*HRAS*	NEG	NEG	NEG
20	FNA	Q96 Pyromark	*HRAS*	NEG	NEG	NEG
21	FNA	Q96 Pyromark	*HRAS*	NEG	NEG	NEG
22	FNA	Q96 Pyromark	*HRAS*	NEG	NEG	NEG
23	FNA	Q96 Pyromark	*HRAS*	NEG	NEG	NEG
24	FFPE	Q96 Pyromark	*HRAS*	NEG	NEG	NEG
25	FNA	Q96 Pyromark	*HRAS*	NEG	NEG	NEG
26	FNA	Q96 Pyromark	*HRAS*	NEG	NEG	NEG
27	FNA	Q96 Pyromark	*HRAS*	NEG	NEG	NEG
28	FNA	Q96 Pyromark	*HRAS*	NEG	NEG	NEG
29	FNA	Q96 Pyromark	*HRAS*	NEG	NEG	NEG
30	FNA	Q96 Pyromark	*HRAS*	NEG	NEG	NEG

**Table 10 diagnostics-15-00390-t010:** Intra-run precision of HRAS mutation detection by Q48 platforms.

Sample	Sample ID	Expected *HRAS* Mutation	Mutation Frequency (%) (Expected)	Detected Mutation (Q48 #1)	Mutation Frequency (%) (Q48 #1)	Detected Mutation (Q48 #2)	Mutation Frequency (%) (Q48 #2)
1	1	NEG	N/A	NEG	N/A	NEG	N/A
	1	NEG	N/A	NEG	N/A	NEG	N/A
	1	NEG	N/A	NEG	N/A	NEG	N/A
2	2	NEG	N/A	NEG	N/A	NEG	N/A
	2	NEG	N/A	NEG	N/A	NEG	N/A
	2	NEG	N/A	NEG	N/A	NEG	N/A
3	3	NEG	N/A	NEG	N/A	NEG	N/A
	3	NEG	N/A	NEG	N/A	NEG	N/A
	3	NEG	N/A	NEG	N/A	NEG	N/A
4	4	Q61K	ND*	Q61K	ND*	Q61K	ND*
	4	Q61K	ND*	Q61K	ND*	Q61K	ND*
	4	Q61K	ND*	Q61K	ND*	Q61K	ND*
5	5	G12D	53	G12D	53	G12D	51
	5	G12D	53	G12D	56	G12D	54
	5	G12D	53	G12D	56	G12D	57
6	6	G12D	41	G12D	39	G12D	40
	6	G12D	41	G12D	42	G12D	40
	6	G12D	41	G12D	38	G12D	40

ND*—not determined; Pyromark programs were not set to calculate peak percentages for this nucleotide change. N/A: Not Applicable as the samples are negative.

**Table 11 diagnostics-15-00390-t011:** Inter-run precision of *HRAS* mutation detection by Q48 platforms.

Run No	Sample ID	Expected *HRAS* Mutation	Mutation Frequency (%) (Expected)	Detected (Q48 #1)	Mutation Frequency (%) (Q48 #1)	Detected (Q48 #2)	Mutation Frequency (%) (Q48 #2)
1	1	G12D	53	G12D	53	G12D	51
2	1	G12D	53	G12D	50	G12D	49
3	1	G12D	53	G12D	47	G12D	47
1	2	G12D	41	G12D	39	G12D	40
2	2	G12D	41	G12D	46	G12D	43
3	2	G12D	41	G12D	49	G12D	49
1	3	NEG	N/A	NEG	N/A	NEG	N/A
2	3	NEG	N/A	NEG	N/A	NEG	N/A
3	3	NEG	N/A	NEG	N/A	NEG	N/A
1	4	NEG	N/A	NEG	N/A	NEG	N/A
2	4	NEG	N/A	NEG	N/A	NEG	N/A
3	4	NEG	N/A	NEG	N/A	NEG	N/A
1	5	NEG	N/A	NEG	N/A	NEG	N/A
2	5	NEG	N/A	NEG	N/A	NEG	N/A
3	5	NEG	N/A	NEG	N/A	NEG	N/A

N/A: Not Applicable as the samples are negative.

**Table 12 diagnostics-15-00390-t012:** Limit-of-detection (LOD) for *HRAS* mutation detection by Q48 platforms.

Sample Dilution Levels	Expected *HRAS* Mutation (%)	Detected Mutation (%) (Q48 #1)	Detected Mutation (%) (Q48 #2)
Undiluted	53	50	49
50%	26.5	29	30
25%	13.25	21	20
12.50%	6.63	9	8
6.25%	3.31	7	8

**Table 13 diagnostics-15-00390-t013:** Performance comparison of pyrosequencing versus NGS for *RAS* mutation detection for thyroid samples.

Sample No.	Sample Type	DNA Conc (ng/µL)	Total Input for NGS (ng)	Original Testing Platform	Gene	Expected Result	Expected Mutation Levels (%)	NGS Result	Detected Mutation %	Concordant (Yes/No)
1	FNA	0.1	2.6	Q48 Pyromark	*KRAS*	G12V	ND*	G12V	11.0%	Yes
					*NRAS*	NEG	N/A	NEG	N/A	
					*HRAS*	NEG	N/A	NEG	N/A	
2	FNA	0.1	2.3	Q48 Pyromark	*KRAS*	G12D	24.0%	G12D	13.4%	Yes
					*NRAS*	NEG	N/A	NEG	N/A	
					*HRAS*	NEG	N/A	NEG	N/A	
3	FNA	0.2	5.2	Q48 Pyromark	*KRAS*	G12D	15.0%	G12D	21.5%	Yes
					*NRAS*	NEG	N/A	NEG	N/A	
					*HRAS*	NEG	N/A	NEG	N/A	
4	FNA	0.3	7.0	Q48 Pyromark	*KRAS*	NEG	N/A	NEG	N/A	Yes
					*NRAS*	Q61R	37.0%	Q61R	38.6%	
					*HRAS*	NEG	N/A	NEG	N/A	
5	FFPE	1.4	29.9	Q48 Pyromark	*KRAS*	NEG	N/A	NEG	N/A	Yes
					*NRAS*	Q61K	ND*	Q61K	44.9%	
					*HRAS*	NEG	N/A	NEG	N/A	
6	FNA	0.8	17.6	Q48 Pyromark	*KRAS*	NEG	N/A	NEG	N/A	Yes
					*NRAS*	Q61R	25.0%	Q61R	25.9%	
					*HRAS*	NEG	N/A	NEG	N/A	
7	FNA	0.2	3.4	Q48 Pyromark	*KRAS*	NEG	N/A	NEG	N/A	Yes
					*NRAS*	Q61R	21.0%	Q61R	18.6%	
					*HRAS*	NEG	N/A	NEG	N/A	
8	FNA	0.1	2.2	Q48 Pyromark	*KRAS*	NEG	N/A	NEG	N/A	Yes
					*NRAS*	NEG	N/A	NEG	N/A	
					*HRAS*	Q61K	ND*	Q61K	28.4%	
9	FNA	12.3	263.5	Q48 Pyromark	*KRAS*	NEG	N/A	NEG	N/A	Yes
					*NRAS*	NEG	N/A	NEG	N/A	
					*HRAS*	Q61K	ND*	Q61K	34.9%	
10	FNA	1.1	24.3	Q48 Pyromark	*KRAS*	NEG	N/A	NEG	N/A	Yes
					*NRAS*	NEG	N/A	NEG	N/A	
					*HRAS*	Q61R	ND*	Q61R	35.9%	
11	FNA	0.1	2.2	Q48 Pyromark	*KRAS*	NEG ALL	N/A	N/A	N/A	Yes
					*NRAS*					
					*HRAS*					
12	FNA	4.4	94.6	Q48 Pyromark	*KRAS*	NEG ALL	N/A	N/A	N/A	Yes
					*NRAS*					
					*HRAS*					
13	FNA	2.4	51.5	Q48 Pyromark	*KRAS*	NEG ALL	N/A	N/A	N/A	Yes
					*NRAS*					
					*HRAS*					
14	FNA	0.1	2.2	Q48 Pyromark	*KRAS*	NEG ALL	N/A	N/A	N/A	Yes
					*NRAS*					
					*HRAS*					
15	FNA	1.2	25.8	Q48 Pyromark	*KRAS*	NEG ALL	N/A	N/A	N/A	Yes
					*NRAS*					
					*HRAS*					
16	FNA	3.3	71.8	Q48 Pyromark	*KRAS*	NEG ALL	N/A	N/A	N/A	Yes
					*NRAS*					
					*HRAS*					
17	FNA	1.6	34.6	Q48 Pyromark	*KRAS*	NEG ALL	N/A	N/A	N/A	Yes
					*NRAS*					
					*HRAS*					
18	FNA	0.1	2.2	Q48 Pyromark	*KRAS*	NEG ALL	N/A	N/A	N/A	#NGS failed
					*NRAS*					
					*HRAS*					
19	FNA	0.1	2.2	Q48 Pyromark	*KRAS*	NEG ALL	N/A	N/A	N/A	NGS failed
					*NRAS*					
					*HRAS*					
20	FFPE	0.1	2.2	Q48 Pyromark	*KRAS*	NEG ALL	N/A	N/A	N/A	#NGS failed
					*NRAS*					
					*HRAS*					
21	FNA	0.3	7.1	Q96 Pyromark	*KRAS*	NEG	N/A	NEG	N/A	Yes
					*NRAS*	G13R	ND*	G13R	17.4%	
					*HRAS*	NEG	N/A	NEG	N/A	
22	FNA	4.6	108.3	Q96 Pyromark	*KRAS*	NEG	N/A	NEG	N/A	Yes
					*NRAS*	Q61R	30.8%	Q61R	24.1%	
					*HRAS*	NEG	N/A	NEG	N/A	
23	FNA	49.2	200.0	Q96 Pyromark	*KRAS*	NEG	N/A	NEG	N/A	Yes
					*NRAS*	Q61R	43.9%	Q61R	42.0%	
					*HRAS*	NEG	N/A	NEG	N/A	
24	FNA	0.6	12.1	Q96 Pyromark	*KRAS*	NEG	N/A	NEG	N/A	Yes
					*NRAS*	Q61R	13.9%	Q61R	9.0%	
					*HRAS*	NEG	N/A	NEG	N/A	
25	FNA	1.7	40.4	Q96 Pyromark	*KRAS*	NEG	N/A	NEG	N/A	Yes
					*NRAS*	Q61K	ND*	Q61K	21.0%	
					*HRAS*	NEG	N/A	NEG	N/A	
26	FNA	0.05	1.1	Q96 Pyromark	*KRAS*	NEG	N/A	NEG	N/A	Yes
					*NRAS*	Q61K	ND*	Q61K	16.8%	
					*HRAS*	NEG	N/A	NEG	N/A	
27	FNA	2.2	52.4	Q96 Pyromark	*KRAS*	NEG	N/A	NEG	N/A	Yes
					*NRAS*	Q61R	45.0%	Q61R	39.3%	
					*HRAS*	NEG	N/A	NEG	N/A	
28	FNA	1.4	33.1	Q96 Pyromark	*KRAS*	NEG	N/A	NEG	N/A	Yes
					*NRAS*	NEG	N/A	NEG	N/A	
					*HRAS*	Q61R	ND*	Q61R	51.2%	
29	FNA	1.6	33.6	Q96 Pyromark	*KRAS*	NEG	N/A	NEG	N/A	Yes
					*NRAS*	NEG	N/A	NEG	N/A	
					*HRAS*	Q61R	ND*	Q61R	28.8%	
30	FNA	0.1	2.4	Q96 Pyromark	*KRAS*	Q61H	40.0	N/A	N/A	#NGS failed
					*NRAS*	NEG	N/A			
					*HRAS*	NEG	N/A			
31	FFPE	1.7	37.2	Q96 Pyromark	*KRAS*	NEG ALL	N/A	Q61K	33.4	No
					*NRAS*			NEG	N/A	
					*HRAS*			NEG	N/A	
32	FNA	1.1	23.7	Q96 Pyromark	*KRAS*	NEG ALL	N/A	NEG	N/A	Yes
					*NRAS*		N/A	NEG		
					*HRAS*		N/A	NEG		
33	FNA	4.0	94.2	Q96 Pyromark	*KRAS*	NEG ALL	N/A	NEG	N/A	Yes
					*NRAS*		N/A	NEG		
					*HRAS*		N/A	NEG		
34	FNA	0.1	2.4	Q96 Pyromark	*KRAS*	NEG	N/A	N/A	N/A	#NGS failed
					*NRAS*	NEG	N/A			
					*HRAS*	Q61K	ND*			
35	FNA	19.8	200.0	Q96 Pyromark	*KRAS*	NEG ALL	N/A	NEG ALL	N/A	Yes
					*NRAS*					
					*HRAS*					
36	FNA	11.7	200.0	Q96 Pyromark	*KRAS*	NEG ALL	N/A	NEG ALL	N/A	Yes
					*NRAS*					
					*HRAS*					
37	FNA	7.3	156.5	Q96 Pyromark	*KRAS*	NEG ALL	N/A	NEG ALL	N/A	Yes
					*NRAS*					
					*HRAS*					
38	FNA	5.2	111.6	Q96 Pyromark	*KRAS*	NEG ALL	N/A	NEG ALL	N/A	Yes
					*NRAS*					
					*HRAS*					
39	FNA	3.0	65.1	Q96 Pyromark	*KRAS*	NEG ALL	N/A	NEG ALL	N/A	Yes
					*NRAS*					
					*HRAS*					
40	FNA	2.2	52.8	Q96 Pyromark	*KRAS*	NEG ALL	N/A	NEG ALL	N/A	Yes
					*NRAS*					
					*HRAS*					
41	FNA	1.5	35.7	Q96 Pyromark	*KRAS*	NEG ALL	N/A	NEG ALL	N/A	Yes
					*NRAS*					
					*HRAS*					
42	FNA	1.2	28.5	Q96 Pyromark	*KRAS*	NEG ALL	N/A	NEG ALL	N/A	Yes
					*NRAS*					
					*HRAS*					
43	FNA	1.1	26.0	Q96 Pyromark	*KRAS*	NEG ALL	N/A	NEG ALL	N/A	Yes
					*NRAS*					
					*HRAS*					
44	FNA	1.0	21.9	Q96 Pyromark	*KRAS*	NEG ALL	N/A	NEG ALL	N/A	Yes
					*NRAS*					
					*HRAS*					
45	FNA	0.1	2.4	Q96 Pyromark	*KRAS*	NEG ALL	N/A	NEG ALL	N/A	Yes
					*NRAS*					
					*HRAS*					
46	FNA	9.1	200.0	Q96 Pyromark	*KRAS*	NEG ALL	N/A	NEG ALL	N/A	Yes
					*NRAS*					
					*HRAS*					
47	FNA	0.5	10.1	Q96 Pyromark	*KRAS*	NEG ALL	N/A	NEG ALL	N/A	Yes
					*NRAS*					
					*HRAS*					
48	FNA	0.1	2.9	Q96 Pyromark	*KRAS*	NEG ALL	N/A	NEG ALL	N/A	Yes
					*NRAS*					
					*HRAS*					
49	FFPE	4.4	103.2	Q96 Pyromark	*KRAS*	NEG ALL	N/A	NEG ALL	N/A	Yes
					*NRAS*					
					*HRAS*					
50	FNA	0.9	19.7	Q96 Pyromark	*KRAS*	NEG ALL	N/A	N/A	N/A	NGS failed
					*NRAS*					
					*HRAS*					
51	FNA	0.1	2.4	Q96 Pyromark	*KRAS*	NEG ALL	N/A	N/A	N/A	#NGS failed
					*NRAS*					
					*HRAS*					
52	FNA	0.1	2.4	Q96 Pyromark	*KRAS*	NEG ALL	N/A	N/A	N/A	#NGS failed
					*NRAS*					
					*HRAS*					
53	FFPE	1.9	40.2	Q96 Pyromark	*KRAS*	NEG ALL	N/A	N/A	N/A	#NGS failed
					*NRAS*					
					*HRAS*					
54	FFPE	0.4	7.8	Q96 Pyromark	*KRAS*	NEG ALL	N/A	N/A	N/A	NGS failed
					*NRAS*					
					*HRAS*					
55	FFPE	0.1	2.4	Q96 Pyromark	*KRAS*	NEG ALL	N/A	N/A	N/A	#NGS failed
					*NRAS*					
					*HRAS*					
56	FFPE	0.1	2.4	Q96 Pyromark	*KRAS*	NEG ALL	N/A	N/A	N/A	#NGS failed
					*NRAS*					
					*HRAS*					

ND*: not determined; Pyromark programs were not set to calculate peak percentages for this nucleotide change. *KRAS* mutation missed by pyrosequencing due to the involvement of the primer binding site. N/A—Not applicable as the samples are negative.

## Data Availability

The original contributions presented in this study are included in the article/Appendix A. Further inquiries can be directed to the corresponding author(s).

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
