# Peer review of "Validation of a Compact and Self-Contained Pyrosequencing Platform for Clinical Screening of *RAS* Mutations in Thyroid Cancers"

_diagnostics, 2025, doi:10.3390/diagnostics15030390_

Round 1

Reviewer 1 Report

Comments and Suggestions for Authors

This paper present a valuable validation of an updated pyrosequencing platform to test for RAS mutations in thyroid and other solid tumors. Given the importance of molecular diagnostics and the expanding fields of personalized medicine, I find the paper is relevant, and deals with an important topic for clinical application. Its novelty lies in the instrument for the mutation analysis, and the validation of the assay on this platform. I have a few minor comments.

Major points:

-        The reference [3] in Li 38 is inaccurately interpreted, or I am missing something from the citation. The paper that was cited reported the highest PPV for HRAS to be 74%. Please clarify the stated PPV values for RAS mutations in IDT samples, or provide another reference (I believe the one under [5] may provide such data).

-        It should be noted in the Introduction that the diagnostic value of RAS mutations in the IDT samples is higher in a panel of mutations, and that RAS mutation testing alone in IDT is not sufficient to influence clinical decision making

-        It would be useful for clarity and structure to add in the Material and Methods section a basic explanation on how the accuracy, precision and LOD of the method were evaluated.

-        Li 95 - which non-thyroid cancers? Were they used as a control?

-        Li 103 - the samples that were tested for comparison with NGS included thyroid samples? This should be more clearly stated, both in the Material and Methods, and throughout the text.

-        Why were different kits for DNA isolation used for thyroid and non-thyroid samples?

-        Was the percentage of DNA background for LLD assessment randomly chosen?

-        Li 235 how were the samples for analytical validations chose? Please provide a short rationale.

-        Li 341 - please provide the measurements in the SI metric system.

-        The Discussion would need to include several more references for the claims in the first and second paragraph.

Grammatical and formatting points:

-        Li 62, Li 215 - missing a full stop at the end of the sentence.

-        Align values in cells in Table 1, Table 9, Table 13

-        Li 308 - justify the text in the paragraph

Reviewer 2 Report

Comments and Suggestions for Authors

The manuscript validates a new, improved, compact Pyrosequencing platform (Q48) for screening RAS mutations in thyroid cancers. This is a valuable area of research, as accurate screening of clinically significant tumor mutations is critical for precision medicine in oncology. here are some additional suggestions for improving the manuscript:

1. The manuscript could benefit from a more detailed comparison of the Q48 platform with other sequencing technologies, particularly in the context of thyroid cancer screening. This could include a discussion on the throughput, cost, and practicality of each method.

2. The discussion could be improved by a more thorough exploration of the limitations of the Q48 platform. This might include a discussion of any technical challenges encountered, potential sources of bias, and how these might impact the results.

3. Given the high failure rate of NGS in certain samples, a more detailed discussion on the reasons behind these failures and potential strategies to mitigate them would be beneficial. This could include a discussion of sample preparation, DNA quality, and sequencing depth.
